# High Flow Oxygen Therapy at Two Initial Flow Settings versus Conventional Oxygen Therapy in Cardiac Surgery Patients with Postextubation Hypoxemia: A Single-Center, Unblinded, Randomized, Controlled Trial

**DOI:** 10.3390/jcm10102079

**Published:** 2021-05-12

**Authors:** Stavros Theologou, Eleni Ischaki, Spyros G. Zakynthinos, Christos Charitos, Nektaria Michopanou, Stratos Patsatzis, Spyros D. Mentzelopoulos

**Affiliations:** 1Department of Cardiac Surgery, Evaggelismos General Hospital, 10675 Athens, Greece; cruxtheol@gmail.com (S.T.); charitosch@otenet.gr (C.C.); nektariamix@hotmail.com (N.M.); patsatzis_stratos@yahoo.gr (S.P.); 2First Department of Intensive Care Medicine, National and Kapodistrian University of Athens Medical School, Evaggelismos General Hospital, 10675 Athens, Greece; eischaki@yahoo.gr (E.I.); szakynthinos@yahoo.com (S.G.Z.)

**Keywords:** high-flow nasal cannula, cardiac surgery, treatment failure, respiratory rate, oxygenation

## Abstract

In cardiac surgery patients with pre-extubation PaO_2_/inspired oxygen fraction (FiO_2_) < 200 mmHg, the possible benefits and optimal level of high-flow nasal cannula (HFNC) support are still unclear; therefore, we compared HFNC support with an initial gas flow of 60 or 40 L/min and conventional oxygen therapy. Ninety nine patients were randomly allocated (respective ratio: 1:1:1) to I = intervention group 1 (HFNC initial flow = 60 L/min, FiO_2_ = 0.6), intervention group 2 (HFNC initial flow = 40 L/min, FiO_2_ = 0.6), or control group (Venturi mask, FiO_2_ = 0.6). The primary outcome was occurrence of treatment failure. The baseline characteristics were similar. The hazard for treatment failure was lower in intervention group 1 vs. control (hazard ratio (HR): 0.11, 95% CI: 0.03–0.34) and intervention group 2 vs. control (HR: 0.30, 95% CI: 0.12–0.77). During follow-up, the probability of peripheral oxygen saturation (SpO_2_) > 92% and respiratory rate within 12–20 breaths/min was 2.4–3.9 times higher in intervention group 1 vs. the other 2 groups. There was no difference in PaO_2_/FiO_2_, patient comfort, intensive care unit or hospital stay, or clinical course complications or adverse events. In hypoxemic cardiac surgery patients, postextubation HFNC with an initial gas flow of 60 or 40 L/min resulted in less frequent treatment failure vs. conventional therapy. The results in terms of SpO_2_/respiratory rate targets favored an initial HFNC flow of 60 L/min.

## 1. Introduction

The high-flow nasal cannula (HFNC) delivers an inspired oxygen fraction (FiO_2_) of 0.21 to 1.0 with a gas flow rate of ≤ 60 L/min [1,2]. FiO_2_ adjustments are independent of flow settings, and patients can receive heated, humidified, and oxygen-rich gas mixtures at flow rates exceeding their own maximum inspiratory flow rates [1,2,3]. HFNC physiological benefits include more predictable FiO_2_ values due to reduced dilution of oxygen [4,5], flow-dependent positive airway pressure [6,7], reduced anatomical dead-space ventilation [8,9], improved mucociliary function and clearance of secretions [10,11], and reduced work of breathing [12]. Therefore, HFNC may improve gas exchange and lung mechanics, reduce the respiratory rate and effort, and ameliorate dyspnea [12,13,14]. Possible HFNC-associated complications include nasal bleeding and mucus dryness, with occasional poor patient tolerance of the device [15,16]. 

Despite the literature-supported physiological benefits, the HFNC’s potential usefulness in postoperative cardiac surgery patients warrants further clarification. More specifically, a meta-analysis of 4 randomized clinical trials (RCTs) of HFNC vs. conventional oxygen therapy after cardiothoracic surgery reported HFNC-associated reductions in the frequency of escalation of respiratory support and pulmonary complications, however no differences in reintubation rate or length of intensive care unit (ICU) and hospital stay [17]. Furthermore, in a more recent meta-analysis, a subgroup analysis of solely cardiac surgery studies failed to confirm any benefit of HFNC with a gas flow of 35–50 L/min [18]; the authors suggested that cardiac surgery increases the risk of postoperative pulmonary complications, and therefore “patients may not benefit from HFNC” [18]. 

Pulmonary complications predisposing sufferers to postextubation hypoxemia (e.g., atelectasis, pneumonia, pleural effusion, and pulmonary edema) may occur in up to 50% of cardiac surgery patients [15,19,20,21,22]. On the other hand, the optimal initial HFNC flow setting for postextubation respiratory support of hypoxemic cardiothoracic surgery patients is still unclear [1,2,23,24]. Data from patients with acute respiratory failure are conflicting; some authors indicate that initial HFNC flows of 35–40 L/min are better tolerated [1,2], whilst others suggest that maximal initial HFNC flows of 60 L/min can rapidly relieve dyspnea, improve oxygenation, and prevent respiratory muscle fatigue [23,24]. A recent physiological randomized crossover study in patients with hypoxemic respiratory failure suggested that the optimal initial HFNC flow may vary according to the “target” respiratory variable (e.g., oxygenation index, minute ventilation, work of breathing, etc.). Therefore, the authors concluded that the initial HFNC flow should be individualized [13].

We hypothesized that (1) HFNC support with a maximal initial flow of 60 L/min or lower flows of 40 L/min might confer benefits relative to conventional oxygen therapy (control) in postoperative cardiac surgery patients with moderate hypoxemia; and (2) HFNC flows of 60 L/min might perform better vs. control relative to flows of 40 L/min.

## 2. Materials and Methods

We conducted a prospective, unblinded RCT in postoperative cardiac surgery patients. The study protocol was approved by the Scientific and Ethics Committee of Evaggelismos Hospital, Athens, Greece (approval no. 47, 3 March 2017). The study was registered 21 days prior to the enrollment of the first patient at clinicaltrials.gov (NCT03282552, registration date 14 September 2017. Principal Investigator: Stavros Theologou). The study was conducted in an eight-bed cardiothoracic ICU. 

The study was conducted in concordance with the Helsinki Declaration [25] and local regulations and ethical standards. Preoperatively, potentially eligible patients or their next-of-kin were informed about the study, both verbally and by pertinent information sheet. Postoperatively, written and informed consent was requested and obtained from the next-of-kin of eligible patients with moderate hypoxemia (see below). Patient consent was requested and obtained as well, as soon as was feasible, depending on their clinical condition [15]. 

### 2.1. Study Population

Inclusion criteria were adult (i.e., age ≥ 18 years) cardiothoracic ICU patients extubated after elective or urgent cardiac surgery. Within 60 min pre-extubation, patients should be alert or oriented, with systolic arterial pressure of 90–160 mmHg and (any) norepinephrine infusion not exceeding 0.15 μg/kg/min. A 60 min spontaneous breathing trial (SBT) was conducted on a T-piece) inspired O_2_ fraction (FiO_2_) = 0.6). Successful SBT fulfilled the following criteria: respiratory rate of 12–29 breaths/min, peripheral oxygen saturation (SpO_2_) > 92%, PaCO_2_ < 45 mmHg, heart rate < 120/min, and systolic pressure or norepinephrine infusion as above. Patients were enrolled if at the end of SBT their PaO_2_/FiO_2_ was < 200 mmHg. Exclusion criteria were obstructive sleep apnea syndrome requiring support with continuous positive airway pressure; preoperative diagnosis of exacerbation of chronic obstructive pulmonary disease; presence of tracheostomy; do not resuscitate status, Glasgow coma scale score < 13; insufficient knowledge of Greek language; and visual or hearing impairment.

### 2.2. Randomization and Study Groups

Following extubation, patients were randomly assigned to intervention groups 1 or 2 or to control group at a ratio of 1:1:1. Blocks of 3 numbers were consecutively drawn from a sequence of 99 unique random numbers (range, 1–99) generated using Research Randomizer version 4.0 (www.randomizer.org, accessed on 2 May 2017). Randomization was performed by the study statistician, who applied the following group allocation rule: the smallest and largest number of each block was assigned to intervention groups 1 and 2, respectively, and the remaining number to control. Upon patient enrollment, attending investigators received an SMS text message containing the patient’s code number and group. 

### 2.3. Interventions and Data Collection

Intervention group 1 received postextubation HFNC oxygen therapy with initial settings of FiO_2_ = 0.6 and gas flow = 60 L/min. Intervention group 2 received HFNC oxygen therapy with initial FiO_2_ = 0.6 and gas flow = 40 L/min. Control patients received oxygen therapy (initial FiO_2_ = 0.6) with a Venturi mask connected to an O_2_ flowmeter set at 15 L/min. For HFNC support, the AIRVO^TM^ 2 instrument (Fisher and Paykel Healthcare, Auckland, New Zealand) with built-in flow generator was used [13] (see Appendix A for additional details).

Patient monitoring included electrocardiographic lead II, intra-arterial pressure, SpO_2_, and respiratory rate. Prespecified data collection time points for SpO_2_, respiratory rate, PaO_2_/FiO_2_, comfort as regards dyspnea and respiratory support modality (visual analogue scale (VAS) score [26]), accessory muscle use, arterial pressure and heart rate, vasopressor support, and core body temperature were within <30 min (baseline) and at 1, 2, and 4 h, then every 4 h onward until 48 h postextubation. Fluid balance was also recorded for the first 24 and 48 h postextubation.

Patients were assessed for downward titration of respiratory support or need for treatment escalation every 4 h postextubation. Gradual weaning from HFNC support included FiO_2_ decrease to 0.5, followed by gas flow decrease to 30 L/min, with a wean-off target of FiO_2_ = 0.4 and gas flow = 20 L/min [6]. If at HFNC FiO_2_ = 0.4 and flow rate = 20 L/min, SpO_2_ and respiratory rate could be respectively maintained at >92% and within 12–20 breaths/min for ≥2 h, patients were switched to the Venturi mask (FiO_2_ = 0.4). In the control group, downward titration of support was aimed at Venturi mask FiO_2_ = 0.4. Patients from all groups fulfilling the aforementioned SpO_2_ or respiratory rate criteria for ≥4 h while on FiO_2_ = 0.4 via Venturi mask were considered for ICU discharge. All patients were scheduled for twice-daily physiotherapy (at 9 am and 7 pm). 

Regarding treatment escalation in intervention groups, if SpO_2_ dropped to ≤92% for ≥5 min at flow rate of <60 L/min, the flow rate was first increased by 5–10 L/min to reduce entrainment of room air during inspiration, increase airway pressure, and recruit alveolar units [2,27]. Subsequently, if SpO_2_ was still ≤92%, FiO_2_ was titrated to SpO_2_ > 92%. In the control group, an SpO_2_ drop to ≤92% for ≥5 min was initially treated with FiO_2_ increase. In all groups, persistent or worsening hypoxemia was ultimately treated with noninvasive ventilation (NIV) or reintubation or invasive ventilation. 

Changes in respiratory support level or modality were ultimately made and approved by the patients’ primary attending physicians. Initiation of mechanical ventilation or ICU discharge within <48 h postextubation resulted in discontinuation of the 4-hourly patient data collection.

### 2.4. Study Outcomes

The primary outcome was the absence of “treatment failure”. Treatment failure was defined as fulfillment of any of the following criteria: (1) any crossover from one assigned treatment to another or change to mechanical ventilatory support; and (2) inability to reverse FiO_2_ or gas flow escalation above initial settings within 48 h of its initiation. Escalation reversal was defined as return to initial (or lower) FiO_2_ or gas flow for ≥4 h.

Secondary outcomes were: (1) maintenance of respiratory rate within 12–20 breaths/min and SpO_2_ > 92%, without escalation of support above initial postextubation level; (2) PaO_2_/FiO_2_; (3) any use of accessory respiratory muscles; and (4) patient comfort and treatment tolerance as assessed by VAS score at specified follow-up time points. Additional outcomes comprised length of ICU and hospital stay, ICU or in-hospital mortality, adverse events (e.g., hypoxemia, need for reintubation, atrial fibrillation, surgical re-exploration due to bleeding, ICU readmission), and any patient discomfort or intolerance related to HFNC. 

### 2.5. Statistical Analysis

A priori power analysis was based on a predicted average treatment failure rate of 15% in intervention groups (10% in intervention group 1 and 20% in intervention group 2) and a failure rate of 51% in the control group. The predicted HFNC-to-control treatment failure ratio of 0.29 corresponded to the lower limit of the 95% confidence interval (CI) of a previously determined “HFNC vs. control” odds ratio for support escalation [28]. For alpha = 0.05 and power = 0.80, 63 patients (*n* = 21 per group) were needed. The selected enrollment of 99 patients (*n* = 33 per group) corresponded to alpha = 0.05 and power = 0.96 and provided a 57% “safety margin” for possible dropouts or missing data. The study’s protocol safety was confirmed by preplanned interim analysis conducted 1 year after the start of the study. 

All analyses were performed according to the intention-to-treat principle. Distribution normality was determined by Kolmogorov–Smirnov test. Continuous variables are presented as the mean ± SD or the median (IQR). Qualitative variables are presented as numbers (percentage). Percentages were compared by Fisher’s exact test.

Regarding the first 48 h postextubation, we assessed (1) SpO_2_ > 92% and a respiratory rate within 12–20 breaths/min as binary outcomes (i.e., maintenance vs. no maintenance of SpO_2_ or respiratory rate above or within the aforementioned limits without escalation of support above initial level) by fitting logistic regression models with group, time, and group* time interaction as explanatory variables; and (2) changes in oxygenation, VAS comfort scale score, and nonoutcome follow-up variables (i.e., PaCO_2_, arterial blood lactate, hemoglobin concentration, hemodynamic variables, temperature, and vasopressor support) by using linear mixed models analyses, fixed factors (group, time, and group* time), a random factor (“patients”), and dependent variables with skewed distributions (e.g., PaO_2_/FiO_2_, PaO_2_, FiO_2_, etc.), which were log-transformed; pairwise comparisons of model estimates were subjected to Bonferroni correction. 

We analyzed the group effect on treatment failure using multivariable Cox regression. Hazard ratios (HRs) and respective 95% CIs were determined for group, European System for Cardiac Operative Risk Evaluation (EuroSCORE) II (which includes age and gender as risk factors), body mass index, cardiopulmonary bypass time, and duration of postoperative sedation and pre-extubation assisted and spontaneous breathing values (see Appendix A for additional details). The reported *p* values are two-tailed. Statistical significance was set at 0.05. Analyses were conducted using SPSS version 25 (IBM corporation, Armonk, NY, USA).

## 3. Results

The study was conducted between 5 October 2017 and 10 May 2019. Of 1174 patients assessed for eligibility, 99 (men, *n* = 67) were enrolled (Figure 1). Patient baseline characteristics were similar (Table 1).

### 3.1. Results for Primary Outcome

Treatment failure occurred in 4/33 (12.1%), 10/33 (30.3%), and 18/33 (54.5%) patients in intervention groups 1 and 2 and control group, respectively. Cox regression revealed lower hazard of treatment failure in intervention group 1 vs. control (HR: 0.11, 95% CI: 0.03–0.34; *p* < 0.001) and intervention group 2 vs. control (HR: 0.30, 95% CI: 0.12–0.77; *p* = 0.012) (Figure 2). There was no significant difference between intervention groups 1 and 2 (HR: 0.35, 95% CI: 0.10–1.26; *p* = 0.11). Body mass index (HR: 1.09, 95% CI: 1.00–1.19; *p* = 0.04) and cardiopulmonary bypass time (HR: 1.01, 95% CI: 1.00–1.01; *p* = 0.03) also predicted treatment failure; see Appendix A for additional Cox model details. 

### 3.2. Results for Secondary Outcomes

Regarding levels of SpO_2_ > 92% and respiratory rate within 12–20 breaths/min without support escalation above the initial level, logistic regression revealed significant group effects on both outcomes (Table 2). At any follow-up time point, intervention group 1 had higher probability of SpO_2_ > 92% and respiratory rate within 12–20 breaths/min compared to the other 2 groups. Accordingly, there were more follow-up time points without support escalation and with SpO_2_ > 92% or respiratory rate within 12–20 breaths/min in intervention group 1 than intervention group 2 or control group (Appendix A Appendix A). Boxplots showing the time courses of SpO_2_ respiratory rates are presented in Appendix A Appendix A.

Table 3 displays the oxygenation results. Regarding PaO_2_/FiO_2_, linear mixed model analysis revealed no significant group effect and a significant effect of group*time; mean estimates for PaO_2_/FiO_2_ increased over time in Intervention groups 1 and 2 and decreased in the control group. There was no significant effect of group or group*time on PaO_2_. There were significant effects of group and group*time on FiO_2_. The estimated marginal means for intervention groups 1 and 2 were lower vs. control. FiO_2_ levels decreased over time in intervention groups 1 and 2 and increased in control. Boxplot presentations of the timecourse of oxygenation are provided in Appendix A Appendix A. The results for the effects of time are reported in Table 3 and Table 4, Appendix A Appendix A, and the respective footnotes.

Table 4 displays results for patient comfort and use of accessory muscles over the first 48 h postextubation. Linear mixed model analysis did not reveal a significant effect of group or group*time on VAS comfort score. The frequency of accessory muscle use did not differ among groups (also see Appendix A Appendix A).

### 3.3. Results on Other Outcomes and Adverse Events

Table 5 displays data and results for the management of treatment failure, other prespecified outcomes, and adverse events or complications. There was less frequent use of nonrebreathing masks (providing FiO_2_~0.9) in intervention groups vs. control. There was no significant between-group difference in escalation to NIV and re-intubation or invasive ventilation. Following attending physician decision, 8 control patients were switched to HFNC with gas flow = 60 L/min and FiO_2_ = 0.9 after being on a nonrebreathing mask for 45.1 ± 3.5 h (range: 36–47 h). The need for any (either reversible or irreversible within 48 h) support escalation due to sustained hypoxemia (i.e., SpO_2_ ≤ 92% for ≥5 min) occurred in less patients of intervention group 1 than intervention group 2; there were no significant differences between intervention groups and control. Data on episodes of hypoxemia in patients without treatment failure are reported in the Appendix A. There were no significant between-group differences in hospital length of stay or clinical course complications. There were no cases of HFNC intolerance. 

### 3.4. Results for Nonoutcome Variables

The results for the rest of the determined nonoutcome variables are detailed in the Appendix A.

## 4. Discussion

This unblinded RCT indicates that HFNC support is superior to conventional oxygen therapy in postoperative cardiac surgery patients with pre-extubation PaO_2_/FiO_2_ < 200 mmHg. The sole difference between protocols for intervention groups 1 and 2 pertained to starting gas flows of 60 and 40 L/min, respectively. At follow-up time points, the probability of support escalation above the initial level due to hypoxemia was 3.2–3.3 times lower in intervention group 1 vs. the other 2 groups. Accordingly, only intervention group 1 consistently performed favorably vs. control as regards levels of SpO_2_ > 92% and respiratory rate within 12–20 breaths/min, without support escalation above the initial level. Measures of oxygenation did not differ significantly among groups; however, administered FiO_2_ was lower in both intervention groups vs. control. Lastly, VAS comfort scores were similar in the 3 groups.

The incidence of at least moderate hypoxemia (i.e., PaO_2_/FiO_2_ < 200 mmHg) upon cardiothoracic ICU admission after on-pump cardiac surgery may amount to ~30% [29]. In a conditional recommendation published in Νovember 2020, HFNC was suggested for postextubation respiratory support of high risk or obese patients undergoing cardiac surgery [30]. HFNC’s physiological benefits are flow-dependent. Increasing the HFNC gas flow from 30 to 60 L/min has been associated with (1) higher positive airway pressures during inspiration and expiration [5], resulting in atelectasis reversal and proportional increase of up to 25% in end expiratory lung volume [31]; (2) more stable FiO_2_ due to reduced entrainment of ambient air [12,32]; and (3) reduced work of breathing and minute ventilation [12,13]. Ventilation becomes more homogenous and the patient’s gas exchange and dyspnea may rapidly improve [12].

In the current study, initial HFNC gas flows of 60 vs. 40 L/min resulted in reduced risk of hypoxemic episodes. Such episodes could be partly attributable to lung recruitment or atelectasis. Indeed, in intervention group 2, desaturation was frequently reversible by increasing the HFNC gas flow up to 60 L/min, thereby likely augmenting the associated positive mean airway and end expiratory pressures and facilitating reopening of collapsed lung units [5,7,12,13,32].

Our inclusion criterion of PaO_2_/FiO_2_ < 200 mmHg at end of SBT was aimed at enrolling the subgroup of patients with concurrent successful SBT and moderate hypoxemia. Randomization of patients before the operation could have resulted in several postrandomization exclusions due to ineligibility (i.e., absence of moderate hypoxemia at end of SBT); such exclusions could have reduced the precision of treatment effect estimates and study power [33]. PaO_2_/FiO_2_ of < 300 mmHg is considered as an independent risk factor for extubation failure [34]. Early postoperative oxygenation disturbances are frequently partly due to atelectasis [35]. However, atelectasis implies a higher potential for lung recruitment [27]. Therefore, we theorized that HFNC gas flows of 40–60 L/min would be more likely to result in lung recruitment without overdistention [27] in this subgroup. Our SpO_2_ results suggest that this was primarily achievable at higher HFNC flows (60 L/min), either in the context of patients being treated as randomized (intervention group 1) or as part of a temporary support escalation process to reverse hypoxemia (intervention group 2).

Our protocol allowed for consecutive titrations of HFNC gas flow or FiO_2_ above their specified initial levels for up to 48 h, before characterizing an ongoing need for such increased support as “treatment failure”. This was in concordance with the projected postoperative occurrence and time course of potentially “persistent atelectasis” (48 h) [36] and of cardiopulmonary bypass-associated, postoperative lung injury (72 to 120 h) [36,37]. The lower probability of treatment failure in intervention group 2 vs. control may partly reflect the effects of frequent temporary support escalation to HFNC gas flows of 50–60 L/min vs. frequent escalation to FiO_2_~0.9 via nonrebreathing mask to treat SpO_2_ ≤ 92%. This is consistent with the similar results for specified SpO_2_ and respiratory rate targets at HFNC gas flows of ≤40 L/min in intervention group 2 vs. control.

Our results are broadly consistent with those of preceding RCTs that employed an inclusion criterion for baseline hypoxemia [15,23]. Vourc’h et al. compared HFNC (45 L/min, FiO_2_ = 1.0) with a Hudson nonrebreathing face mask in 90 cardiac surgery patients with postextubation SpO_2_ < 96% at FiO_2_ = 0.5 via Venturi mask [15]. The results for HFNC vs. control included higher PaO_2_/FiO_2_ (by 22–26%) at 6 and 24 h, lower respiratory rate (by 14%) at 24 h, improved patient tolerance and satisfaction, and decreased NIV use; there were no differences in reintubation rate or ICU mortality. Maggiore et al. studied HFNC (50 L/min) vs. Venturi mask treatments in 105 general ICU patients with postextubation PaO_2_/FiO_2_ < 300 mmHg and reported improvements in oxygenation, respiratory rate, and patient comfort; less episodes of interface displacement and desaturation; and less frequent need for NIV or reintubation in the HFNC group, with no differences in ICU mortality [23]. The current study’s paucity of HFNC-associated, significant PaO_2_/FiO_2_ or VAS comfort score improvements may be partly attributable to the switch of 8/33 control patients (24.2%) to HFNC. Furthermore, given their low incidence, the current study was likely underpowered to detect differences in reintubation and mortality rates.

Regarding the inconclusive results of the recent meta-analyses [17,18], the absence of hypoxemia-based inclusion criteria in the meta-analyzed RCTs [28,38,39,40] may have reduced the likelihood of detecting a benefit of HFNC; furthermore, the enrollment of patients with a body mass index of ≥ 30 kg/m^2^ in one study [38] may have hampered the capability of atelectasis reversal via the 3–4 cm HFNC end expiratory pressure.

In the current study, ICU and hospital length of stay were not affected by HFNC support. This contrasts with a recently reported, HFNC-associated 29% reduction in hospital length of stay in elective cardiac surgery patients [41]. This discrepancy may be due to differences in eligibility criteria, as the study of Zochios et al. included only patients with pre-existing respiratory disease [41].

The current study’s strengths include its RCT design and high follow-up rates, as well as the protocolized use of 2 initial HFNC gas flow levels at FiO_2_ = 0.6, followed by downward titrations of HFNC settings according to specified physiological targets. To our knowledge, this is the first concurrent evaluation of such HFNC protocols vs. conventional oxygen therapy in a selected subgroup of hypoxemic cardiac surgery patients with high risk of extubation failure [34].

Randomization of patients in a single intervention group and a control group (at a ratio of 2:1) would not have enabled randomized comparisons of HFNC protocols using different starting gas flows (either vs. control or between intervention groups). The results of such comparisons did reveal clinically relevant between-group differences in the secondary outcome (Table 2 and Appendix A of the Appendix A). In routine practice, decreases in SpO_2_ and increases in respiratory rate frequently guide changes in HFNC gas flow settings.

The study limitations included the single-center, unblinded design and the relatively small sample size of hypoxemic cardiac surgery patients, which might have led to overestimation of treatment effects [42]. It is uncertain whether the current results can be generalizable to other subpopulations of critically ill patients, such as those with ventilator-induced diaphragmatic dysfunction and postextubation respiratory failure [15,43]. Combined therapies such as NIV and HFNC [44] may be more preferable to NIV or HFNC alone [45] for general ICU patients with high risk of extubation failure. Furthermore, the frequent support escalation in the control group and intervention group 2 may have confounded the evaluation of patient comfort. Lastly, we did not determine the occurrence of mucus dryness [15].

In conclusion, in this single-center, unblinded RCT of hypoxemic cardiac surgery patients, postextubation HFNC with an initial gas flow of 60 or 40 L/min resulted in less frequent treatment failure compared to conventional oxygen therapy. The results for specified SpO_2_ and respiratory rate targets favored the use of an initial HFNC gas flow of 60 L/min rather than 40 L/min.

## Figures and Tables

**Figure 1 jcm-10-02079-f001:**
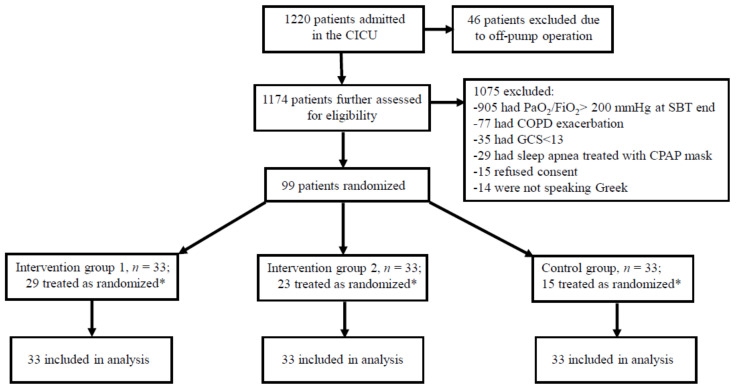
The study flow chart. Treated as randomized is defined as treated according to study protocol, without fulfilling the prespecified criteria for treatment failure as detailed in the Methods. CICU, cardiothoracic intensive care unit; SBT, spontaneous breathing trial; COPD, chronic obstructive pulmonary disease; CPAP, continuous positive airways pressure; PaO_2_/FiO_2_, arterial oxygen partial pressure to inspired oxygen fraction ratio; GCS, Glasgow Coma Score. * The remaining patients in each group (intervention group 1, *n* = 4; intervention group 2, *n* = 10; control group, *n* = 18) received escalation of respiratory support in the context of fulfilling predefined criteria for treatment failure (also see Methods).

**Figure 2 jcm-10-02079-f002:**
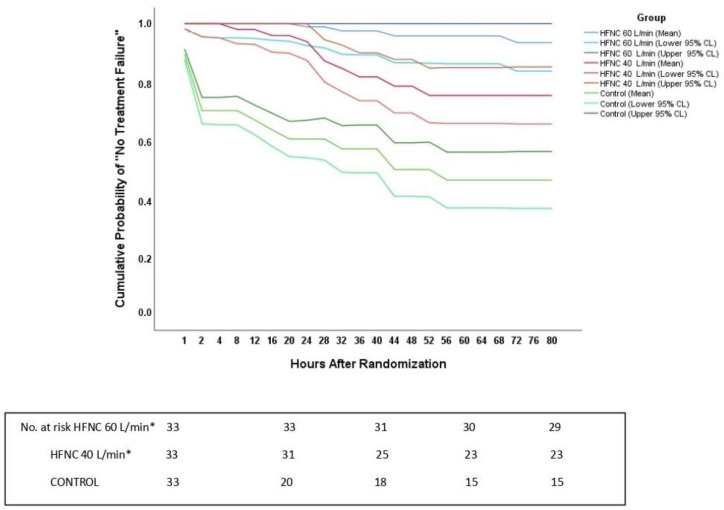
Cumulative probability (mean, 95% confidence interval) of the absence of treatment failure (i.e., “no treatment failure”) in intervention group 1 (HFNC 60 L/min), intervention group 2 (HFNC 40 L/min), and control group. HFNC, high-flow nasal canula. Cox model covariates: group, European System for Cardiac Operative Risk Evaluation (EuroSCORE) II (which includes age and gender as risk factors), body mass index, cardiopulmonary bypass time, duration of postoperative sedation, and duration of pre-extubation assisted and spontaneous breathing values. Collinearity diagnostics: condition index: 22.0; variance inflation index: 1.07–1.36. CL, confidence limit. * Initial HFNC gas flow level.

**Table 1 jcm-10-02079-t001:** Demographics and baseline clinical characteristics for the three study groups.

Scheme 1.	Intervention 1, HFNC 60 L/min * (Ν = 33)	Intervention 2, HNFC 40 L/min * (Ν = 33)	Control (Ν = 33)
Age (years), mean ± SD	65.7 ± 10.5	67.0 ± 9.1	68.6 ± 7.5
Male sex (%)	23 (69.7)	22 (66.7)	22 (66.7)
BMI (kg/m^2^), mean±SD	28.9 ± 5.9	29.0 ± 5.0	29.8 ± 3.7
EuroSCORE II, median (IQR)	2.3 (1.1–3.5)	2.3 (1.3–3.9)	1.9 (1.3–3.2)
CPB time (min), median (IQR)	116 (100–154)	119 (98–176)	108 (83–145)
Ischemia time (min), median (IQR)	70 (57–102)	71 (56–86)	65 (44–89)
Operation Type			
CABG, no. (%)	14 (42.4)	16 (48.5)	17 (51.5)
Valve replacement ^†^, no. (%)	10 (30.3)	7 (21.2)	8 (24.2)
Aortic valve/ascending aorta and/or aortic arch replacement, no. (%)	6 (18.2)	8 (24.2)	6 (18.2)
CABG and valve replacement, no. (%)	3 (9.1)	2 (6.1)	2 (6.1)
Postoperative CMV Settings ^‡^			
FiO_2_ median (IQR)	0.6 (0.5–0.6)	0.6 (0.5–0.6)	0.5 (0.5–0.6)
PEEP (cmH_2_O) median (IQR)	8 (6–8)	8 (6–8)	6 (6–8)
Tidal volume (mL/kg PBW ^§^), mean±SD	7.9 ± 0.7	7.9 ± 0.7	8.0 ± 0.6
End-of-Operation to Extubation			
Sedation time ICU (hours), median (IQR)	6.5 (4.5–14.0)	11.5 (4.8–18.5)	5.5 (3.8–10.0)
Duration of Intubation (hours), median (IQR)	12.5 (6.8–20.5)	19.0 (12.0–36.5)	12.0 (7.0–20.0)
Time on PSV + SBT duration (min) **, median (IQR)	180 (120–240)	180 (120–435)	240 (150–420)
Pre-extubation, SBT PaO_2_/FiO_2_ (mmHg)	144.2 ± 24.3	148.1 ± 26.3	156.3 ± 29.1
Presumed Etiology of Hypoxemia			
Atelectasis, no. (%)	18 (54.5)	21 (63.6)	21 (63.6)
Cardiogenic pulmonary edema, no. (%)	3 (9.1)	3 (9.1)	3 (9.1)
Pneumonia, no. (%)	6 (18.2)	4 (12.1)	3 (9.1)
CPB-associated lung injury, no. (%)	6 (18.2)	5 (15.2)	6 (18.2)
Physiological Data and Vasopressor Support upon Study Enrollment ^††^			
SpO_2_ (%), mean ± SD	95.9 ± 2.7	96.3 ± 2.3	97.1 ± 1.6
PaO_2_/FiO_2_ (mmHg) mean ± SD	135.1 ± 37.2	145.4 ± 51.2	171.6 ± 55.5
PaCO_2_ (mmHg), mean ± SD	41.3 ± 5.4	41.8 ± 4.7	42.2 ± 5.2
Arterial pH, mean ± SD	7.39 ± 0.05	7.39 ± 0.05	7.37 ± 0.05
Arterial blood lactate (mmol/L), mean ± SD	2.1 ± 1.0	1.7 ± 0.7	1.9 ± 1.1
Hemoglobin concentration (g/dL), mean ± SD	11.2 ± 1.9	10.3 ± 1.8	10.6 ± 1.3
Mean arterial pressure (mmHg), mean ± SD	79.8 ± 11.5	83.1 ± 11.5	77.8 ± 6.7
Heart rate (beats/min), mean ± SD	90.3 ± 12.7	91.4 ± 16.5	88.2 ± 12.6
Core body temperature (degrees Celsius), mean ± SD	36.9 ± 0.6	37.1 ± 0.6	37.0 ± 0.5
Norepinephrine IR (μg/kg/min), median (IQR)	0.02 (0.00–0.06)	0.00 (0.00–0.06)	0.04 (0.00–0.07)

HFNC, high flow nasal cannula; BMI, body mass index; EuroSCORE II, European System for Cardiac Operative Risk Evaluation; CPB, cardiopulmonary bypass; CABG, coronary artery bypass grafting; CMV, controlled mechanical ventilation; FiO_2_, inspired oxygen fraction; PEEP, positive end expiratory pressure; PBW, predicted body weight; PSV, pressure support ventilation; SBT, spontaneous breathing trial; SpO_2_, peripheral oxygen saturation; PaO_2_, oxygen arterial partial pressure; IR, infusion rate. * Initial HFNC gas flow level. ^†^ Replacement of the aortic, mitral, or tricuspid valve. ^‡^ Respiratory rate was titrated to an arterial pH of > 7.30. ^§^ Calculated as 0.9 × (height (cm) − 150) + 50.0 kg in male patients, and as 0.9 × (height (cm) − 150) + 45.5 kg in female patients. ** Time on PSV coincides with the pre-extubation period of assisted breathing, whereas time on SBT coincides with the pre-extubation period of spontaneous breathing. ^††^ Just prior to study protocol initiation in all groups; at this time point, all study participants were on conventional oxygen therapy.

**Table 2 jcm-10-02079-t002:** Results of logistic regression analyses for levels of SpO_2_ > 92 and respiratory rate within 12 to 20 breaths/min, without escalation of support above its initial level.

	OR	95% CI	*p*-Value
SpO_2_ > 92%			
Group			
Intervention 1 (HFNC 60 L/min *) vs. control	3.17	(2.14–4.67)	<0.001
Intervention 2 (HFNC 40 L/min *) vs. control	0.93	(0.65–1.33)	0.69
Intervention 1 vs. Intervention 2	3.26	(2.25–4.76)	<0.001
Time	0.99	(0.98–0.99)	<0.001
Interaction Group * time			
Intervention 1 vs. control	1.00	(0.98–1.01)	0.76
Intervention 2 vs. control	0.99	(0.98–1.01)	0.25
Intervention 1 vs. Intervention 2	1.01	(0.99–1.02)	0.25
Respiratory Rate within 12 to 20 breaths/min			
Group			
Intervention 1 vs. control	2.37	(1.65–3.41)	<0.001
Intervention 2 vs. control	1.02	(0.71–1.47)	0.91
Intervention 1 vs. Intervention 2	1.93	(1.34–2.79)	<0.001
Time	0.98	(0.98–0.99)	<0.001
Interaction Group * time			
Intervention 1 vs. control	0.99	(0.98–1.00)	0.12
Intervention 2 vs. control	0.99	(0.98–1.00)	0.15
Intervention 1 vs. Intervention 2	1.01	(1.00–1.02)	0.15

OR, odds ratio; CI, confidence interval; SpO_2_, peripheral oxygen saturation; HFNC, high-flow nasal canula. * Initial HFNC gas flow level.

**Table 3 jcm-10-02079-t003:** Results of mixed-model analyses for PaO_2_/FiO_2_, PaO_2_, and FiO_2_.

Dependent Variable—PaO_2_/FiO_2_	F	*p*-Value	AIC	% Var.
Effect of Group (fixed factor)	2.3	0.10	−1792.7	66.9%
Effect of Time (fixed factor)	1.1	0.34		
Effect of Group * Time (interaction)	1.6	0.048		
Group—Pairwise Comparisons—PaO_2_/FiO_2_	Estimated Marginal Mean		95% CI	
			Lower	Upper
Intervention 1 (HFNC 60 L/min *)-mmHg	152.5		136.9	169.8
Intervention 2 (HFNC 40 L/min *)-mmHg	148.4		133.3	165.2
Control-mmHg	130.7		117.4	145.5
Dependent Variable—PaO_2_	F	*p*-Value	AIC	% Var.
Effect of Group (Fixed Factor)	0.8	0.48	−2437.9	46.4%
Effect of Time (Fixed Factor)	2.6	0.002 †		
Effect of Group * Time (Interaction)	0.8	0.70		
Group—Pairwise Comparisons—PaO_2_	Estimated Marginal Mean		95% CI	
			Lower	Upper
Intervention 1 (HFNC 60 L/min *)-mmHg	84.8		80.3	89.5
Intervention 2 (HFNC 40 L/min *)-mmHg	87.0		82.4	91.8
Control-mmHg	88.9		84.2	93.8
Dependent variable—FiO_2_	F	*p*-Value	AIC	% Var.
Effect of Group (Fixed Factor)	9.1	<0.001	−2578.4	61.4%
Effect of Time (Fixed Factor)	1.9	0.03 ‡		
Effect of Group * Time (Interaction)	2.1	0.003		
Group—Pairwise Comparisons—FiO_2_	Estimated Marginal Mean		95% CI	
			Lower	Upper
Intervention 1 (HFNC 60 L/min *)	0.55 ^§^		0.52	0.59
Intervention 2 (HFNC 40 L/min *)	0.58 **		0.54	0.62
Control-mmHg	0.68		0.63	0.72

F, value of the F statistic for the effects of the fixed factors and of their interaction; PaO_2_, oxygen arterial partial pressure; FiO_2_, inspired oxygen fraction; AIC, Akaike’s information criterion for goodness of fit; % Var., percent variance (of the observed values) explained by the linear mixed model estimates; CI, confidence interval; HFNC, high-flow nasal canula. Log transformation of oxygenation data (also see statistical analysis) was reversed for the purpose of numeric presentation. * Initial HFNC gas flow level. † Bonferroni-corrected pairwise comparisons revealed that the mean estimate for PaO2 of the total study population was significantly lower at 20, 32, and 40 h relative to 4 h after extubation. ‡ Bonferroni-corrected pairwise comparisons of overall mean estimates for FiO2 at postextubation follow-up time points did not reveal any significant difference; ^§^, *p* < 0.001 vs. control; **, *p* = 0.007 vs. Control.

**Table 4 jcm-10-02079-t004:** Results of mixed-model analysis for the VAS comfort scale score and of the pairwise comparisons for the frequency of use of accessory respiratory muscles at the follow-up time points.

Dependent Variable—VAS Score	F	*p*-Value	AIC	% Var.
Effect of Group (Fixed Factor)	2.6	0.08	−4407.6	67.2%
Effect of Time (Fixed Factor)	5.2	<0.001 ^†^		
Effect of Group * Time (Interaction)	1.2	0.28		
Group—Pairwise Comparisons—VAS Score	Estimated Marginal Mean		95% CI	
			Lower	Upper
Intervention 1 (HFNC 60 L/min *)	7.9		7.6	8.2
Intervention 2 (HFNC 40 L/min *)	7.6		7.3	7.9
Control	7.5		7.2	7.7
Use of Accessory Muscles	No. (%) of Follow-Up Time Points within Each Group			*p*-Value
Intervention 1 (HFNC 60 L/min *) vs. Control	14 (3.5%) vs. 10 (2.3%)			0.41 ^‡^
Intervention 2 (HFNC 40 L/min *) vs. Control	16 (4.0%) vs. 10 (2.3%)			0.23 ^‡^
Intervention 1 vs. Intervention 2	14 (3.5%) vs. 16 (4.0%)			0.72 ^‡^

F, value of the F statistic for the effects of the fixed factors and of their interaction; VAS, visual analogue scale; AIC, Akaike’s information criterion for goodness of fit; % Var., percent variance (of the observed values) explained by the linear mixed model estimates; CI, confidence interval; HFNC, high-flow nasal canula. Logarithmic transformation of the VAS scores (also see statistical analysis) was reversed for the purpose of numeric presentation. * Initial HFNC gas flow level. ^†^ Bonferroni-corrected pairwise comparisons revealed that the mean estimates for VAS score of the total study population exhibited significant improvements at ≥8 h relative to ≤4 h postextubation. ^‡^ Value not corrected for multiple comparisons.

**Table 5 jcm-10-02079-t005:** Management of treatment failure, “other” prespecified outcomes, and adverse events.

	Group					
	Intervention 1 (HFNC 60 L/min *; *n* = 33)	Intervention 2 (HFNC 40 L/min *; *n* = 33)	Control (*n* = 33)	*p*-Value	*p*-Value	*p*-Value
				1 vs. 2	2 vs. 3	1 vs. 3
Nonrebreathing Mask in Patients with Treatment Failure, No. (%)	2 (6.1)	5 (15.2)	15 (45.5) †	0.43	0.045 ^‡^	<0.001 ^‡^
NIMV following treatment failure, No. (%)	0 (0.0)	3 (9.1) ^§^	1 (3.0)	0.24	0.61	>0.99
Intubation/IMV following treatment failure, No. (%)	2 (6.1)	5 (15.2)	2 (6.1)	0.43	0.43	>0.99
Discomfort causing discontinuation of treatment, No (%)	0 (0.0)	0 (0.0)	0 (0.0)	-	-	-
Length of CICU stay (hours), median (IQR)	53.0 (32.0–77.5)	65.0 (39.5–76.5)	55.0 (35.0–70.0)	0.44	0.29	0.97
Length of hospital stay (days), median (IQR)	9.0 (7.0–12.0)	8.0 (6.5–10.5)	7.0 (6.0–9.5)	0.61	0.29	0.12
Adverse Events						
Need for any support escalation due to sustained hypoxemia, No. (%) **	17 (51.5)	28 (84.8)	23 (69.7)	<0.02 ^‡^	0.24	0.21
Need for transfusion (packed red blood cells) in the CICU, No. (%)	11 (33.3)	14 (42.4)	15 (45.5)	0.61	>0.99	0.45
Delirium in the CICU, No. (%)	8 (24.2)	13 (39.4)	4 (12.1)	0.29	0.07	0.34
Atrial fibrillation in the CICU, No. (%)	6 (18.2)	12 (36.4)	6 (18.2)	0.17	0.17	>0.99
Post-discharge readmission to the CICU (for any indication)	4 (12.1)	4 (12.1)	2 (6.1)	>0.99	0.67	0.67
Cardiac Arrest / died in the CICU, No. (%)/No. (%) Cardiac Arrest / died in-hospital after CICU discharge No. (%)/No. (%)	2 (6.1)/1 (3.0) 4 (12.1)/4 (12.1)	3 (9.1)/2 (6.1) 2 (6.1)/2 (6.1)	1 (3.0)/1 (3.0) 0 (0.0)/0 (0.0)	>0.99/>0.99 0.67/0.67	0.61/>0.99 0.49/0.49	>0.99/>0.99 0.11/0.11
Acute Renal Failure in the CICU, No. (%)	2 (6.1)	2 (6.1)	0 (0.0)	>0.99	0.49	0.49
Surgical re-exploration due to bleeding in the CICU, No. (%) Surgical re-exploration due to bleeding after CICU discharge No. (%)	0 (0.0) 0 (0.0)	1 (3.0) 2 (6.1)	0 (0.0) 0 (0.0)	>0.99 0.49	>0.99 0.49	>0.99 >0.99
Pneumothorax in the CICU, No. (%)	1 (3.0)	2 (6.1)	0 (0.0)	>0.99	0.49	>0.99
Epileptic seizures in the CICU, No. (%)	0 (0.0)	2 (6.1)	0 (0.0)	0.49	0.49	-
Chest wound infection during hospital stay, No. (%)	0 (0.0)	0 (0.0)	0 (0.0)	-	-	-

HFNC, high-flow nasal canula; CICU, cardiothoracic intensive care unit; NIMV, noninvasive mechanical ventilation; IMV, invasive mechanical ventilation. Adverse events reported to have occurred “solely” in the CICU were not observed after CICU discharge. * Initial HFNC gas flow level. † Eight patients were crossed over to HFNC with a gas flow of 60 L/min and an FiO_2_ of 0.8 after 45.1 ± 3.5 of breathing with a nonrebreathing mask. ^‡^ The original *p*-Value of the corresponding pairwise comparison was subjected to the Bonferroni correction for 3 comparisons (i.e., multiplied by 3); other presented and originally nonsignificant *p*-Values for the 3 pairwise comparisons were not subjected to any correction. ^§^ All 3 patients were subsequently re-intubated for IMV. ** Any sustained hypoxemia-related escalation of respiratory support (i.e., increase in HFNC gas flow or inspired oxygen fraction, or initiation of noninvasive or invasive mechanical ventilation), either reversible or not reversible within 48 h (also see Methods); sustained hypoxemia was defined as a drop in peripheral oxygen saturation to ≤92% for at least 5 min.

## Data Availability

De-identified datasets used or analyzed during the current study are available (in the form of Microsoft Excel Worksheets) from the corresponding author on reasonable request. Surrogates or patients who provided informed consent for study participation also granted permission to the sharing of de-identified study data with persons authorized by the principal investigator and first author.

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
