# Peer review of "High Flow Oxygen Therapy at Two Initial Flow Settings versus Conventional Oxygen Therapy in Cardiac Surgery Patients with Postextubation Hypoxemia: A Single-Center, Unblinded, Randomized, Controlled Trial"

_jcm, 2021, doi:10.3390/jcm10102079_

Round 1

Reviewer 1 Report

The authors present results of an interesting prospective, unblinded clinical RCT about compearing two inithial setting HFNC vs conventional oxygen in cardiac surgery patients with postextubation hypoxemia.

In my opinion this is very well-designed study with good primary and secondary outcomes and clear results. 

However, I think that it is worth to develope the introduction to make it more interesting in the beginning. Authors can add  few words about high-flow therapy, taking into account the indications, contraindications, benefits and complications, especially in patients after cardiac surgery.

Moreover, there are no information how often is a problem of postextubation hypoxemia after cardiac surgery. Authors could add this information to show, how important it is to find the resolve of this problem. 

Author Response

We would like to thank you for your effort to help us improve our paper. Please note that revision associated changes are highlighted in red script (in the text of the revised submission).

Reviewer 1

Comments and Suggestions for Authors

1] The authors present results of an interesting prospective, unblinded clinical RCT about comparing two initial setting HFNC vs conventional oxygen in cardiac surgery patients with postextubation hypoxemia.

In my opinion this is very well-designed study with good primary and secondary outcomes and clear results. 

Response: Thank you for your positive comments.

2] However, I think that it is worth to develop the introduction to make it more interesting in the beginning. Authors can add few words about high-flow therapy, taking into account the indications, contraindications, benefits and complications, especially in patients after cardiac surgery.

Response: Thank you, we followed your suggestion. Accordingly, the first 2 paragraphs of the Introduction now read as follows:

The high-flow nasal cannula (HFNC) delivers an inspired oxygen fraction (FiO2) of 0.21 to 1.0 with a gas flow rate of ≤60 L/min [1, 2]. FiO2 adjustments are independent of flow settings, and patients can receive heated, humidified and oxygen-rich gas mixtures at flow rates exceeding their own maximum inspiratory flow rates [1-3]. HFNC physiological benefits include more predictable FiO2 values due to reduced dilution of oxygen [4, 5], flow-dependent positive airway pressure [6, 7], reduced anatomical dead-space ventilation [8, 9], improved mucociliary function and clearance of secretions [10, 11] and reduced work of breathing [12]. Therefore, HFNC may improve gas-exchange and lung mechanics, reduce respiratory rate and effort, and ameliorate dyspnea [12-14]. Possible HFNC-associated complications include nasal bleeding and mucus dryness, with occasional poor patient tolerance of the device [15, 16]. 

Despite the literature-supported physiological benefits, HFNC potential usefulness in postoperative cardiac surgery patients warrants further clarification. More specifically, a meta-analysis of 4 randomized clinical trials (RCTs) of high-flow nasal cannula (HFNC) vs. conventional oxygen therapy after cardiothoracic surgery reported HFNC-associated reductions in the frequency of escalation of respiratory support and pulmonary complications, but no difference in reintubation rate, and length of intensive care unit (ICU) and hospital stay [17]. However Furthermore, in a more recent meta-analysis, a subgroup analysis of solely cardiac surgery studies failed to confirm any benefit of HFNC with gas flow of 35-50 L/min [18];  meta-analysis authors suggested that cardiac surgery per se increases the risk of postoperative pulmonary complications and therefore ″patients may not benefit from HFNC″ [18].″

Moreover, there are no information how often is a problem of postextubation hypoxemia after cardiac surgery. Authors could add this information to show, how important it is to find the resolve of this problem. 

Response: Thank you. The third paragraph of the Introduction now reads as follows:

Pulmonary complications predisposing to postextubation hypoxemia (e.g. atelectasis, pneumonia, pleural effusion, and pulmonary edema) may occur in up to 50% of cardiac surgery patients [15, 19-22]. On the other hand, the optimal initial HFNC flow setting for postextubation respiratory support of hypoxemic cardiothoracic surgery patients is still unclear [1, 2, 23, 24]. Data from patients with acute respiratory failure are conflicting; some authors indicate that initial HFNC flows of 35–40 L/min are better tolerated [1, 2], whilst others suggest that maximal initial HFNC flows of 60 L/min can rapidly relieve dyspnea, improve oxygenation and prevent respiratory muscle fatigue [23, 24]. A recent physiologic randomized crossover study in patients with hypoxemic respiratory failure suggested that optimal initial HFNC flow may vary according to ″target″ respiratory variable (e.g. oxygenation index, minute ventilation, work of breathing, etc.). Therefore, the authors concluded that initial HFNC flow should be individualized [13].

Accordingly, we have added 12 supporting references to the reference list of the revised paper. Please note that the newly added references are highlighted in red script (Iin the reference list).

Reviewer 2 Report

This is a timely randomized clinical trial comparing 3 groups of treatments with a randomization 1:1:1.

Group 1: high flow nasal cannula intervention with 60 l/min and FiO2 0.6

Group 2:high flow nasal cannula intervention with 40 l/min and FiO2 0.6

Control: conventional O2 flow with Venturi mask 15 l/min

I have some minor comments:

I would randomize only 2 groups, Intervention and control group. I think that randomization of 3 groups may generate confusion among the readers. Moreover, no differences were reported between the two intervention groups.

Please, clarify the reason why randomization was done after extubation instead of before the operation.

Please, add CI band in the Kaplan-Meire curves.

Author Response

We would like to thank you for your effort to help us improve our paper. Please note that revision associated changes are highlighted in red script (in the text of the revised submission).

Reviewer 2

Comments and Suggestions for Authors

This is a timely randomized clinical trial comparing 3 groups of treatments with a randomization 1:1:1.

Group 1: high flow nasal cannula intervention with 60 l/min and FiO2 0.6

Group 2: high flow nasal cannula intervention with 40 l/min and FiO2 0.6

Control: conventional O2 flow with Venturi mask 15 l/min

I have some minor comments:

1] I would randomize only 2 groups, Intervention and control group. I think that randomization of 3 groups may generate confusion among the readers. Moreover, no differences were reported between the two intervention groups.

Response: Thank you. Randomization of patients in a single Intervention group and a control group (e.g. at a ratio of 2:1) would not have enabled randomized comparisons of HFNC protocols using different starting gas flows (either vs. control or between Intervention groups). Although, high-flow nasal cannula (HFNC) groups did not differ significantly as regards treatment failure, they did differ with respect to the achievement of the pre-specified SpO2 and respiratory rate targets (please see Table 2 and Table S2 of the Supplement). Such differences are clinically relevant in our opinion, as in our routine practice, changes in HFNC gas flow settings are frequently guided by drops in SpO2 and increases in respiratory rate.  

Accordingly, we have added the following text as fifth paragraph of page 13 of the revised manuscript:

    ″Randomization of patients in a single Intervention group and a control group (at a ratio of 2:1) would not have enabled randomized comparisons of HFNC protocols using different starting gas flows (either vs. control or between Intervention groups). Results of such comparisons did reveal clinically relevant between-group differences on the secondary outcome (Table 2, and Table S2 of the Supplement). In our routine practice, drops in SpO2 and increases in respiratory rate frequently guide changes in HFNC gas flow settings.″    

2] Please, clarify the reason why randomization was done after extubation instead of before the operation.

Response: Thank you. In response to your comment, we have added the following text to paragraph 4 of page 12 of the revised manuscript:

    ″Our inclusion criterion of PaO2/FiO2<200 mmHg at SBT’s end was aimed at enrolling the subgroup of patients with concurrent successful SBT and moderate hypoxemia. Randomization of patients before the operation could have resulted in several postrandomization exclusions due to ineligibility (i.e. absence of moderate hypoxemia at SBT’s end); such exclusions could have reduced the precision of treatment effect estimates and study power [33]. ″

Accordingly, we have added reference 33 to the reference list of the revised manuscript. This reference analyzes issues associated with postrandomization exclusion (which did not occur in the present study).

3] Please, add CI band in the Kaplan-Meire curves.

Response: Thank you. Please note that the “Analyze – Survival – Cox Regression” function of SPSS version 26 does not enable the generation of plots with 95% confidence intervals of constructed survival probability curves. Therefore, we used information from the “Survival Table” of the Cox Regression output, in order to produce the requested graph. More specifically, the “Survival Table” reports the standard error (SE) of the mean probability of survival at each time point of follow-up. Accordingly, we computed the datapoints of the upper and lower confidence limit lines by multiplying the SE by 1.96 and (respectively) adding and subtracting this product from the (known) mean probability (of treatment failure). Subsequently, we used the “Graph – Legacy Dialogues – Line – Multiple” function of SPSS version 26 to construct the “mean probability (95% confidence interval)l” lines for the 3 study groups. This resulted in the revised Figure 2, which is "embedded" in page 7 of the revised submission.

Please note that datapoints corresponding to a probability of >1.0 were not plotted.

Also, please note that the revised Figure legend now reads as follows:

Figure 2. Cumulative probability (mean, 95% confidence interval) of the absence of treatment failure, i.e. ″no treatment failure″ in Intervention group 1 (HFNC 60 L/min), Intervention group 2 (HFNC 40 L/min), and Control group. HFNC, high-flow nasal canula. Cox model covariates: Group, European System for Cardiac Operative Risk Evaluation (EuroSCORE) II (which includes age and gender as risk factors), body mass index, cardiopulmonary bypass time, duration of postoperative sedation, and duration of pre-extubation assisted and spontaneous breathing. Collinearity diagnostics: Condition index: 22.0; Variance inflation index: 1.07-1.36. CL, confidence limit. *, Reflects initial HFNC gas flow level.

We do hope that the quality of the revised Figure will be deemed acceptable.